# Electrocardiogram Abnormality Detection Using Machine Learning on Summary Data and Biometric Features

**DOI:** 10.3390/diagnostics15070903

**Published:** 2025-04-01

**Authors:** Kennette James Basco, Alana Singh, Daniel Nasef, Christina Hartnett, Michael Ruane, Jason Tagliarino, Michael Nizich, Milan Toma

**Affiliations:** 1Department of Computer Science, College of Engineering and Computing Sciences, New York Institute of Technology, 1855 Broadway, New York, NY 10023, USA; kmaddela@nyit.edu; 2Entrepreneurship and Technology Innovation Center, College of Engineering and Computing Sciences, New York Institute of Technology, Old Westbury, NY 11568, USA; asing195@nyit.edu; 3Department of Osteopathic Manipulative Medicine, College of Osteopathic Medicine, New York Institute of Technology, Old Westbury, NY 11568, USA; dnasef02@nyit.edu; 4Catholic Health Service of Long Island, 245 Old Country Rd, Melville, NY 11747, USA; christina.hartnett@chsli.org (C.H.); michael.ruane@chsli.org (M.R.); jason.tagliarino@chsli.org (J.T.)

**Keywords:** ECG/EKG, ECG-related biometrics, Extremely randomized trees, gradient boosted trees, support vector machines

## Abstract

**Background/Objectives:** Electrocardiogram data are widely used to diagnose cardiovascular diseases, a leading cause of death globally. Traditional interpretation methods are manual, time-consuming, and prone to error. Machine learning offers a promising alternative for automating the classification of electrocardiogram abnormalities. This study explores the use of machine learning models to classify electrocardiogram abnormalities using a dataset that combines clinical features (e.g., age, weight, smoking status) with key electrocardiogram measurements, without relying on time-series data. **Methods:** The dataset included demographic and electrocardiogram-related biometric data. Preprocessing steps addressed class imbalance, outliers, feature scaling, and the encoding of categorical variables. Five machine learning models—Gaussian Naive Bayes, support vector machines, random forest trees, extremely randomized trees, gradient boosted trees, and an ensemble of top-performing classifiers—were trained and optimized using stratified k-fold cross-validation. Model performance was evaluated on a reserved testing set using metrics such as accuracy, precision, recall, and F1-score. **Results:** The extremely randomized trees model achieved the best performance, with a testing accuracy of 66.79%, recall of 66.79%, and F1-score of 62.93%. Ventricular rate, QRS duration, and QTC (Bezet) were identified as the most important features. Challenges in classifying borderline cases were noted due to class imbalance and overlapping features. **Conclusions:** This study demonstrates the potential of machine learning models, particularly extremely randomized trees, in classifying electrocardiogram abnormalities using demographic and biometric data. While promising, the absence of time-series data limits diagnostic accuracy. Future work incorporating time-series signals and advanced deep learning techniques could further improve performance and clinical relevance.

## 1. Introduction

Electrocardiogram (ECG) data plays a critical role in diagnosing cardiovascular diseases, which are a leading cause of mortality worldwide [1]. Traditional ECG interpretation relies on trained medical professionals to visually inspect the ECG waveforms and identify abnormalities [2]. However, this manual process can be time-consuming, subjective, and prone to human error [2,3,4]. In recent years, machine learning (ML) techniques have shown great promise in automating ECG diagnosis, enabling more efficient and accurate detection of cardiovascular conditions [5,6,7]. However, existing studies primarily rely on time-series ECG data, which, while highly feature-rich, may not always be available in all clinical settings. This study focuses on the practical application of ML models in scenarios where access to such time-series signals is unavailable. This reflects real-world challenges faced by many healthcare facilities, particularly those with limited resources or infrastructure, where only summary ECG data and demographic features are accessible. By demonstrating the feasibility of using limited data for ECG abnormality classification, our work highlights the potential of ML to extend diagnostic support to a broader range of clinical environments. Such approaches are particularly valuable in resource-constrained contexts, where even incremental improvements in clinical workflows can have significant impacts on patient care.

The features of ECG biometrics are derived from the characteristic waveforms of the heart’s electrical activity, including the P-R interval, QRS duration, Q-T interval, and axes such as P, R, and T. These features are influenced by physiological and demographic factors such as age, weight, and heart rate, making them highly individualized. ECG-related biometrics are explored not only for their diagnostic potential but also for their utility in classification tasks using ML algorithms. By combining clinical features (e.g., age, weight, and smoking status) with key ECG measurements, this study aims to classify ECG abnormalities without relying on time-series data.

Recent advancements in ML have advanced ECG analysis by improving diagnostic precision and interpretability through novel architectures and optimization techniques [8]. A foundational review underscores the transformative role of AI in automating cardiovascular disease diagnosis, particularly through deep learning models that enhance accuracy and efficiency [9]. Building on this, innovative architectures like xECGArch leverage both short- and long-term ECG features to achieve high interpretability and performance in atrial fibrillation detection [10], while portable models such as CardioAttentionNet combine BiLSTM networks and attention mechanisms for robust arrhythmia classification [11]. Feature engineering also plays a critical role, as evidenced by metaheuristic algorithms that optimize feature selection to attain near-perfect accuracy in arrhythmia classification [12]. Despite these advancements, challenges like data scarcity and class imbalance persist, prompting the systematic exploration of data augmentation methods to improve model generalizability under constrained datasets [13]. Together, these studies highlight a growing emphasis on hybrid methodologies that integrate architectural innovation, interpretability, and data-centric approaches, setting the stage for clinically adaptable solutions in ECG abnormality detection.

This study investigates the use of various shallow ML models, including extremely randomized trees (EXT), gradient boosted trees (GBT), support vector machines (SVM), and others, to classify a subject’s ECG status based on a unique dataset containing both clinical features (e.g., age, weight, heart rate) and key ECG measurements (such as the P-R interval and QRS duration). Using this combination of demographic and ECG-related biometric data, trained models aim to accurately identify ECG abnormalities without relying on the actual ECG time series signal.

The study addresses several challenges throughout the research process, including exploratory data analysis, class imbalance, data preprocessing, model selection, and hyperparameter tuning. Class imbalance, in particular, posed a significant challenge, as the majority of samples were labeled as “Abnormal ECG”, which could lead to biased models that struggle to correctly classify minority classes. To mitigate this, techniques such as stratified sampling and the use of evaluation metrics like the F1-score were employed. The methodology for this study involved preprocessing the dataset to handle outliers, scaling features, and encoding categorical variables. Multiple ML models, including Gaussian Naive Bayes, support vector machines, random forest trees, extremely randomized trees, and gradient boosted trees, were trained and optimized using stratified k-fold cross-validation. Model performance was evaluated on a reserved testing set using metrics such as accuracy, precision, recall, and F1-score. The study also explored feature importance and reframed the classification problem to address challenges in distinguishing borderline ECG cases.

The contribution of this study lies in applying the foundational steps of ML (ranging from data analysis, preprocessing, training, and hyperparameter optimization) to address the challenges inherent to working with an incomplete dataset: specifically, the absence of time-series ECG signals, the presence of outlier values, and a relatively small sample size present significant obstacles. However, this study demonstrates that clinically useful and realistic results can still be achieved by focusing on real-world scenarios where access to complete datasets, such as time-series ECG signals, may not always be possible. By addressing these challenges in a methodical and transparent manner, the study highlights the potential of ML models to provide value in resource-constrained settings without inflating performance expectations.

## 2. Methods

This section details the processes of data preparation, analysis, and ML model development. The data collection process is described, including the sources and types of data utilized. Exploratory data analysis is conducted to examine the dataset’s structure, distributions, and potential issues such as outliers and imbalances. Data preprocessing steps are outlined, including cleaning, transforming, and preparing the data for ML, with a focus on handling missing values, scaling features, and addressing class imbalance. The section also explains the training and optimization of ML models, covering the selection of algorithms, hyperparameter tuning, and evaluation methods to ensure robust and reliable performance.

### 2.1. Data Collection

An anonymized dataset of *N* = 4466 ECG summary records was acquired for this study from a regional healthcare system in the New York metro area. The dataset included a diverse range of demographic, clinical, and biometric features, such as age, weight, smoking status, ventricular rate, atrial rate, and key ECG measurements (e.g., P-R interval, QRS duration, Q-T interval, and QTC (Bezet)). To ensure patient privacy, all personally identifiable information was removed prior to data access, and the dataset was provided in compliance with applicable data protection regulations. The data represented a heterogeneous population, reflecting the variability in patient characteristics and cardiovascular health statuses. This diversity was critical for developing and evaluating machine learning models capable of generalizing across different patient populations. However, the dataset also presented challenges, such as class imbalance, with a majority of samples labeled as “Abnormal ECG”, and the presence of outliers, such as implausible weight or heart rate values, likely resulting from data entry errors. These issues were addressed during preprocessing to ensure the dataset was suitable for training robust and reliable machine learning models.

### 2.2. Exploratory Data Analysis

Data visualizations and analysis of summary statistics, e.g., the feature mean and its distribution, presence of outliers, etc., reveal additional information about the data [14]. This guided the authors on what data preprocessing methods are necessary or relevant for each feature, such as value encoding, normalizing/standardization to some distribution, outlier handling, etc.

Table 1 shows some examples of the dataset (after the removal of nuisance columns such as IDs that are unique to each observation—which does not provide predictive value).

Figure 1 shows the target distribution, i.e., the frequency of value occurrences of the DIAGNOSIS_LINE. The majority of the data has Abnormal ECG values, indicating an imbalanced dataset. This imbalance is addressed in Section 2.3 Data Preprocessing via stratified sampling.

The distributions of the numerical features are shown in Figure 2. A cursory visual analysis of the researchers leads to the conclusion that most of the features follow a Gaussian distribution, while some have a long right-tailed distribution.

The outlier analysis of the numerical columns WEIGHT, VENTRICULAR RATE, ATRIAL RATE is shown in Figure 3. Outlier values above 95% of the cumulative distribution function are expected to be realistic, e.g., subjects with obesity or poor cardiovascular health. However, outliers on the lower end are unrealistic and are considered absurd outlier values, e.g., a person having a weight value of less than 50 lbs or heart rate of 10 beats/min: This may be due to some flaw that occurred in the data collection process of the source. Again, this issue is addressed in Section 2.3.

### 2.3. Data Preprocessing

In this subsection, we focus on the significant preprocessing steps performed such as target value curation, treatment of columns that had values originating from non-standardized data collection, and handling of categorical, ordinal, and numerical features. Some examples were preprocessed (The examples in Table 2 are not necessarily the same examples from Table 1).

#### 2.3.1. Target Creation

Target features were created by consolidating the multi-line format of the DIAGNOSIS_LINE_* columns into one column (DIAGNOSIS) via concatenation, separated by a space. Then, by case-insensitive string matching via the substring operator ∋, target values of the observations containing the keys outlined in the map in Equation (Equation 1) are replaced with their corresponding ordinal value:(1)$$\forall d\in \mathrm{DIAGNOSIS},\;\;f\left(d\right)=\left\{\begin{array}{cc} 0, & \mathrm{if}\;“\mathrm{Normal}\;\mathrm{ECG}”∋d\\ 1, & \mathrm{else}\;\mathrm{if}\;“\mathrm{Borderline}\;\mathrm{ECG}”∋d\\ 2, & \mathrm{else}\;\mathrm{if}\;“\mathrm{Abnormal}\;\mathrm{ECG}”∋d \end{array}\right.$$

Finally, those observations that did not contain any of the keys were dropped from the dataset.

#### 2.3.2. Smoker Column

Redundant and synonymous smoker values, e.g., Some Days and Light Smoker, were consolidated. This column is coded as an ordinal variable ranging from observations that individuals did not smoke to those that they had heavy smoke exposure. The re-labeling of every possible smoker value into its corresponding ordinal is outlined in the map in Equation (Equation 2):(2)$$\forall s\in \mathrm{SMOKING}\_\mathrm{STATUS},\;\;f\left(s\right)=\left\{\begin{array}{cc} 0, & \mathrm{if}\;“\mathrm{never}”∋s\\ 1, & \mathrm{if}\;“\mathrm{never}\;\mathrm{assessed}”∋s\vee “\mathrm{unknown}”∋s\\ 2, & \mathrm{if}\;“\mathrm{former}”∋s\\ 3, & \mathrm{if}\;“\mathrm{smoker},\;\mathrm{current}\;\mathrm{status}\;\mathrm{unknown}”∋s\\ \; & \;\vee “\mathrm{light}\;\mathrm{smoker}”∋s\\ \; & \;\vee “\mathrm{some}\;\mathrm{days}”∋s\\ \; & \;\vee “\mathrm{passive}\;\mathrm{smoke}\;\mathrm{exposure}\;-\;\mathrm{never}\;\mathrm{smoker}”∋s\\ 4, & \mathrm{if}\;“\mathrm{every}\;\mathrm{day}”∋s\vee “\mathrm{heavy}\;\mathrm{smoker}”∋s \end{array}\right.$$

#### 2.3.3. Other Nominal Columns

Other nominal columns, SEX and DIABETES, are encoded as 0s and 1s. N.B., the DIABETES column may be considered as a categorical feature, i.e., having no order in the values, since it only has binary values. Finally, examples with missing values of these features are imputed using their mode.

#### 2.3.4. Numerical Columns

The remaining numerical columns, including those with long right-tailed distributions, were scaled to a standard normal distribution (the choice to standardize long-tailed distributions, rather than apply a log transformation or min-max scaling, was guided by cross-validation results), with careful attention to outlier values. To address these outliers, these features are scaled into a robust standard normal that involved removal of the median and scaling of the data based on the interquartile range [15]. Standardizing these columns also offers the added advantage of improving the training efficiency of gradient-based methods and SVMs, which are particularly sensitive to data scaling [16]. Finally, examples with missing values of these features are imputed using their mean.

#### 2.3.5. Dataset Split

Permutation of the dataset was performed—see the Source Code and Scripts section that guarantees deterministic/reproducible results. Then, the dataset was split into an 80:20 ratio, where 80% ($N_{\mathrm{train}}=3188$) of the data is for the training set and the other 20% ($N_{\mathrm{test}}=798$) is for the test set. A further 80:20 stratified cross-validation split on the training set produced a partial-training set and validation set for the purposes of hyperpameter optimization.

### 2.4. Model Training and Hyperparameter Tuning

#### 2.4.1. The Baseline (Null) Classifier

Given the multiclass classification at hand (3 classes), a naive or null model would predict any sample using the dataset’s target mode, which is DIAGNOSIS = “Abnormal ECG” or 2, which comprises (52.36%) majority of the full dataset. That is, a higher accuracy than that will be a successful classifier.

Considering the practicality of the model (specifically, its value in predicting patient ECG data) and the unbalanced nature of the dataset, the primary metric of interest will be the F1 score. This metric, which is the harmonic mean of precision and recall, is robust to class imbalance [17].

#### 2.4.2. Model Training and Optimization

Classifiers based on different priors are considered, e.g., tree-based, gradient-based, etc., to enable various exploration of the unknown dataset hypothesis. All classification models trained are as follows: K-nearest neighbor (KNN), Gaussian Naive Bayes (GNB) [18], support vector machine (SVM) [19], random forest (RFT) [20], extremely randomized trees (EXT) [21], and gradient boosted trees (GBT) [22], Finally, an ensemble of the top 4 fine-tuned weak learners (ENS) is used, leveraging a meta-learning approach. This ensemble predicts classes using a soft-voting scheme that averages the predictions and confidence scores of the individual weak learners. Soft voting, in contrast to majority (hard) voting, helps handle ties in the overall predictions. Equation (Equation 3) represents the soft-voting mechanism, where the final prediction $\hat{y}$ for a given diagnosis *d* is determined by selecting the class *c* that maximizes the weighted sum of probabilities $P_{m}\left(c\right)$ predicted by each weak learner *m*, with weights $w_{m}$. This approach incorporates the confidence of each model, making it more robust than hard voting.(3)$$\forall d\in \mathrm{DIAGNOSIS},\;\hat{y}=\arg\max_{c\in C}\sum_{m=1}^{M}w_{m}P_{m}\left(c\right)$$

Model variety and the No Free Lunch Theorem guided the researchers to explore and choose the best performing model on different metrics [23,24,25].

Stratified k-fold (K = 5) validation was used to train the models to ensure robustness in validation metrics, achieve partial train-validation split, and avoid any bottleneck of the lowest frequency class.

A grid search of plausible hyperparameters for every model is explored, e.g., number of neighbors for each sample, depth for tree-based methods, penalty parameter C for SVM, etc., again using stratified k-fold validation. This allows for the hyperparameter exploration of each model to avoid overfitting the training data, i.e., for the purposes of model regularization.

Finally, all models are parameterized with their best hyperparameters and retrained using the entire training dataset, that is, including the validation set, to maximize the available data for learning and enhance the model’s generalization performance [23].

## 3. Results

The results of this study are presented in the following subsections. First, the performance of the hyperparameter-optimized models is evaluated using the reserved testing set, with particular emphasis on the SVM model, which outperformed most models across several metrics. Although the ensemble of multiple classifiers (SVM, EXT, GNB, and GBT) showed superior performance compared to the individual SVC model, the authors selected the most parsimonious model based on Occam’s razor. The confusion matrix analysis provides a detailed view of the model’s classification accuracy and highlights challenges in distinguishing borderline ECG cases. To address these challenges, the problem is reframed as a binary classification task and the trade-offs between precision and recall are explored using a precision–recall curve. Finally, feature importance analysis identifies the most influential predictors that contribute to the model performance, providing valuable information on the clinical and biometric parameters critical to ECG classification.

### 3.1. Model Evaluation on the Testing Set

The hyperparameter-optimized models were evaluated using the reserved test set. Table 3 summarizes the test metrics for all models, highlighting the best-performing metrics for each. The results indicate that the extremely randomized trees (EXT) model consistently outperformed all other models in most metrics of interest, including testing accuracy, recall, and F1 score. Figure 4 shows a visualization of the results in Table 3.

The computational efficiency of the models was assessed during the training and evaluation phases. All models were trained and tested on standard computational hardware, with training times ranging from a few seconds to several minutes depending on the model complexity and hyperparameter tuning. Prediction times for new data were near-instantaneous, demonstrating the practicality of the approach for real-world clinical applications. The overall computational requirements are minimal, making the proposed solution feasible for deployment in resource-constrained settings.

### 3.2. Overview of the SVM Model

The SVM model demonstrated superior performance [19]. For the given dataset, the optimal hyperparameters for the SVM model were determined to be a tolerance C of 1 and using the radial basis function as the kernel. These hyperparameters were identified through grid search and cross-validation, ensuring the model’s robustness and ability to generalize effectively.

### 3.3. Confusion Matrix Analysis

Figure 5 presents the confusion matrix for the EXT model’s predictions on the testing set. The matrix highlights the distribution of true versus predicted classes for normal, borderline, and abnormal ECG diagnoses.

The confusion matrix reveals that the EXT model struggled to classify borderline ECG cases accurately. These cases were often misclassified as either normal or abnormal ECGs, reflecting the inherent difficulty in distinguishing borderline cases due to their overlap with the other two categories.

### 3.4. Binary Classification Reframing

A naive solution to address the high and dangerous cost of a false negative diagnosis of a patient (and also false positives, depending on the situation and goal) is to reframe the problem from multiclass to binary classification by forming an equivalence between the target values: Borderline ECG ≡ Abnormal ECG. The modified target map in Equation (Equation 4) replaces the DIAGNOSIS values with its corresponding ordinals:(4)$$\forall d\in \mathrm{DIAGNOSIS},\;\;f\left(d\right)=\left\{\begin{array}{cc} 0, & \mathrm{if}\;“\mathrm{Normal}\;\mathrm{ECG}”∋d\\ 1, & \mathrm{else}\;\mathrm{if}\;“\mathrm{Borderline}\;\mathrm{ECG}”∋d\;\vee \;“\mathrm{Abnormal}\;\mathrm{ECG}”∋d \end{array}\right.$$

This conversion from multiclass to binary classification allows the model to have a precision–recall (PR) trade-off curve with a dynamic threshold, enabling the selection of a specific precision value along with its corresponding recall, and vice versa [26,27,28].

Figure 6 shows the PR curve of the EXT on the testing set with 95% target values for both precision and recall. Depending on the situation and goal, to minimize either false negatives or false positives, the decision threshold for prediction can be adjusted to 0.7967 (for 95% precision) and 0.3883 (for 95% recall), respectively. This reframing allowed for the generation of a precision–recall (PR) curve, as shown in Figure 6. The PR curve demonstrates the trade-off between precision and recall at varying decision thresholds, enabling the selection of thresholds based on specific clinical priorities.

Key thresholds were identified, including one achieving 95% precision (with recall at 33.21%) and another achieving 95% recall (with precision at 68.09%). These thresholds provide flexibility for optimizing the model based on the desired balance between minimizing false positives and minimizing false negatives, depending on the clinical context.

### 3.5. Feature Importance Analysis via the EXT

Although EXT did not outperform SVC, it is still valuable to know the decision function made by this bagging-trees model (EXT outperformed RFT). Feature importance analysis for the EXT model revealed that ventricular rate, QRS duration, and QTC (Bezet) were the most influential features in predicting ECG classifications. Figure 7 illustrates the relative importance of each feature, along with the variability in importance scores across different subsets of the data.

These findings highlight the critical role of specific clinical and biometric parameters in distinguishing among normal, borderline, and abnormal ECG statuses. The variability in feature importance scores underscores the robustness of these key predictors across different subsets of the data.

## 4. Discussion

The implications of these misclassifications are critical in a healthcare context. False negatives, where borderline ECGs are misclassified as normal, may lead to complacency in patients, delaying necessary medical interventions. Conversely, false positives, where borderline ECGs are misclassified as abnormal, could result in unnecessary treatments, causing undue stress, financial burden, and potential harm to patients. These risks underscore the importance of minimizing both false negatives and false positives in medical diagnostic models to ensure accurate and actionable predictions.

### 4.1. Challenges in Classifying Borderline ECG Cases

The classification of borderline ECG cases presented a significant challenge in this study. These cases, which lie between normal and abnormal ECGs in terms of risk, were the most difficult to classify accurately. This difficulty is likely due to the inherent overlap in features between the borderline category and the other two classes. From a clinical perspective, this overlap reflects the continuum of cardiovascular health, where borderline cases may share characteristics with both normal and abnormal ECGs. Consequently, the model’s misclassifications often involved predicting borderline cases as either normal or abnormal.

The values in the multiple grouped distributions represent different clinical and biometric features (e.g., ventricular rate, QRS duration, weight, etc.) and their respective frequencies or measurements across the dataset. These distributions are important because they provide insights into the variability and patterns within the data, which directly influence the performance of the machine learning models used to classify ECG outcomes. For example, the distributions reveal key challenges such as class imbalance (e.g., the majority of samples being labeled as “Abnormal ECG”) and the presence of outliers (e.g., unrealistic weights or heart rates). These issues must be addressed during preprocessing to ensure the models are trained on clean, representative data. Additionally, the differences in distributions highlight the heterogeneity of the dataset, which reflects the diversity of patient characteristics and conditions.

Relating these distributions back to the use of ML in evaluating ECG outcomes, they underscore the complexity of the classification task. AI models, particularly machine learning algorithms, rely on patterns in the data to make predictions. Understanding the distributions helps identify which features are most informative for distinguishing between “Normal”, “Borderline”, and “Abnormal” ECGs. For instance, features like ventricular rate and QRS duration, which exhibit distinct patterns in their distributions, were found to be among the most important predictors in the study. These distributions are critical as they guide the preprocessing steps (e.g., scaling, outlier removal, and stratified sampling) and influence model selection and optimization. By addressing the challenges revealed in the distributions, the study ensures that the AI models are robust and capable of accurately classifying ECG abnormalities, ultimately improving diagnostic efficiency and reducing the risk of human error in clinical practice.

### 4.2. Reframing the Problem as Binary Classification

To address the challenges associated with borderline ECG classification, the problem was reframed as a binary classification task. By combining the borderline and abnormal ECG categories into a single “at-risk” class, the model’s focus shifted to distinguishing between normal and at-risk cases. This reframing allowed for the generation of a precision–recall (PR) curve, which provided a dynamic threshold for optimizing the trade-off between precision and recall.

The PR curve enabled the identification of thresholds tailored to specific clinical priorities. For instance, a threshold achieving high precision minimizes false positives, ensuring that patients classified as at-risk are highly likely to require medical attention. Conversely, a threshold achieving high recall minimizes false negatives, ensuring that nearly all at-risk patients are identified. This flexibility is particularly valuable in medical applications, where the cost of misclassification varies depending on the clinical context. For example, in screening scenarios, high recall may be prioritized to ensure no at-risk patients are missed, while in treatment planning, high precision may be preferred to avoid unnecessary interventions.

### 4.3. Feature Importance and Clinical Relevance

The feature importance analysis of the extremely randomized trees (EXT) model provided valuable insights into the predictors most influential in ECG classification. Ventricular rate, QRS duration, and QTC (Bezet) were identified as the top contributors to the model’s predictions. These features are well-established indicators of cardiovascular health, with abnormalities in these parameters often associated with arrhythmias, conduction disorders, and other cardiac conditions. The robustness of these features across different subsets of the data highlights their clinical relevance and supports their use in automated diagnostic models.

The variability in feature importance scores, as indicated by the standard deviations across out-of-bag samples, suggests that while these features are consistently influential, their relative contributions may vary depending on the specific characteristics of the dataset [12]. This variability underscores the importance of incorporating diverse and representative data in model training to ensure generalizability across different patient populations.

### 4.4. Limitations and Future Directions

Despite the promising results, this study faced several limitations that warrant further investigation. First, the dataset contained unrealistic outliers, such as subjects with weights below 50 lbs or heart rates below 10 beats per minute. These anomalies likely resulted from errors during data collection and were addressed during preprocessing. However, a cleaner and more standardized dataset would likely improve model performance and reliability.

Second, the absence of time-series ECG signal data limited the scope of analysis to static demographic and biometric features. ECG data are inherently time-series in nature, and the inclusion of raw signal data could enable the application of advanced deep learning techniques, such as long short-term memory (LSTM) networks or convolutional neural networks (CNNs). These methods are well-suited for capturing temporal patterns and could significantly enhance the model’s diagnostic accuracy. Future studies should prioritize access to time-series ECG data to explore these possibilities.

Finally, the class imbalance in the dataset posed a challenge, particularly for the borderline ECG category. While stratified sampling and evaluation metrics such as the F1-score were employed to mitigate this issue, alternative approaches, such as synthetic data generation or cost-sensitive learning, could further improve the model’s ability to handle imbalanced datasets.

Hence, to improve the diagnostic capabilities and practical application of ML models for ECG abnormality detection, future research should prioritize several key directions. First, incorporating time-series ECG signal data into the analysis is essential, as such data captures temporal patterns critical for accurate diagnosis. Other advanced deep learning techniques could be leveraged to process these time-series signals to enhance model performance [10]. Additionally, efforts should be made to obtain larger and more diverse datasets that reflect a wide range of patient demographics and clinical conditions, thereby improving the generalizability of the models. Another critical area for exploration is the development of explainable AI (XAI) systems, which can provide clinicians with interpretable insights into the decision-making process, fostering greater trust and adoption in clinical settings. Lastly, standardized validation frameworks, such as those aligned with regulatory benchmarks like the FDA guidance [29], should be adopted to ensure robust model evaluation and facilitate meaningful cross-study comparisons.

Furthermore, field-programmable gate array (FPGA)-accelerated architectures achieve 8–10× higher energy efficiency than GPUs through reduced-bitwidth computations [30]. These hardware optimizations could enhance model portability in resource-constrained clinical settings where low latency and energy efficiency are critical [31]. Future work should investigate dedicated hardware implementations, such as FPGA-based inference accelerators, to improve computational throughput for real-time deployment. Techniques like 8-bit quantization [32] could reduce model size and processing latency without compromising diagnostic accuracy, aligning with energy-efficient edge computing advancements.

### 4.5. Implications for Clinical Practice

The findings of this study demonstrate the potential of ML models, particularly the EXT model, in automating the classification of ECG abnormalities using demographic and biometric data. The ability to dynamically adjust decision thresholds based on clinical priorities offers a practical advantage, allowing healthcare providers to tailor the predictions of the model to specific diagnostic or screening scenarios [33]. However, the challenges associated with borderline ECG classification highlight the need for careful consideration of the clinical context and the potential consequences of misclassification. Either way, as of now, no ML model is intended to be used exclusively for medical diagnosis. These models are designed to serve as supplementary tools to assist physicians by providing potential insights or predictions, rather than making definitive clinical decisions. Their primary purpose is to guide clinicians on what to expect and to support, not replace, human expertise.

Hence, while the EXT model showed promise in classifying ECG abnormalities, further improvements in data quality, feature representation, and model architecture are necessary to fully realize the potential of machine learning in this domain. Future research should focus on addressing these limitations and exploring the integration of time-series data and advanced deep learning techniques to enhance diagnostic accuracy and clinical utility.

### 4.6. Challenges in Cross-Study Comparability

A significant challenge in the field of ML for healthcare care, particularly in studies such as this, is the lack of standardized validation frameworks, which makes comparisons between studies of model performance unreliable. Although some studies report impressively high accuracy metrics, these often result from inadequate validation practices, such as improper data splitting, a lack of external testing, or overfitting to specific datasets. This issue is particularly relevant when comparing our rigorously validated accuracy of 66.79% with reported in other studies that claim higher but potentially less reliable metrics.

The absence of unified methods for evaluating ML models leads to inflated and misleading performance claims. Many studies do not adopt robust practices such as three-way data splitting (training, validation, and testing) or external validation on independent datasets. Without these practices, reported metrics may reflect overfitting or data leakage rather than true generalizability. For example, a model achieving 98% accuracy on a poorly validated dataset may appear superior to a rigorously validated model with lower accuracy, but its clinical utility is practically meaningless.

In addition, the variability in the datasets, preprocessing techniques, and evaluation metrics between studies further complicates the comparisons. Factors such as differences in patient demographics, data collection methods, and feature engineering can significantly influence model performance. Without standardized reporting and validation protocols, it becomes impossible to discern whether a model’s reported accuracy reflects genuine clinical utility or methodological artifacts.

In this study, rigorous validation practices were prioritized, including stratified k-fold cross-validation and careful handling of class imbalance, to ensure the reliability of the results. Although our accuracy may appear modest compared to some studies, it reflects a realistic and reproducible performance level, free from overfitting or data leakage. Until the field adopts standardized validation frameworks, such as those outlined in recent FDA guidance [29], comparing ML studies remains speculative and potentially misleading. Therefore, we caution against equating high reported accuracies with clinical reliability and emphasize the importance of robust validation in evaluating ML models.

## 5. Conclusions

This study demonstrates the potential of ML models in detecting abnormalities in ECGs using summary ECG data and biometric features. By addressing the challenges of missing time-series ECG signals, outliers, and class imbalance in a relatively small dataset, the research highlights the feasibility of applying ML in real-world, resource-constrained settings. The EXT model emerged as the most effective classifier, achieving the highest accuracy and recall, with ventricular rate, QRS duration, and QTC (Bezet) identified as critical predictive features. The results underscore the practicality of leveraging summary ECG and biometric data for ECG abnormality detection, making this approach particularly suitable for healthcare environments where access to time-series data is limited. This study’s findings demonstrate deploying ML-based diagnostic tools as scalable solutions in diverse clinical environments. Future work could explore refining the models using larger datasets, incorporating additional biometric parameters, and validating the models in varied clinical contexts to ensure broader applicability and improved diagnostic outcomes.

## Figures and Tables

**Figure 1 diagnostics-15-00903-f001:**
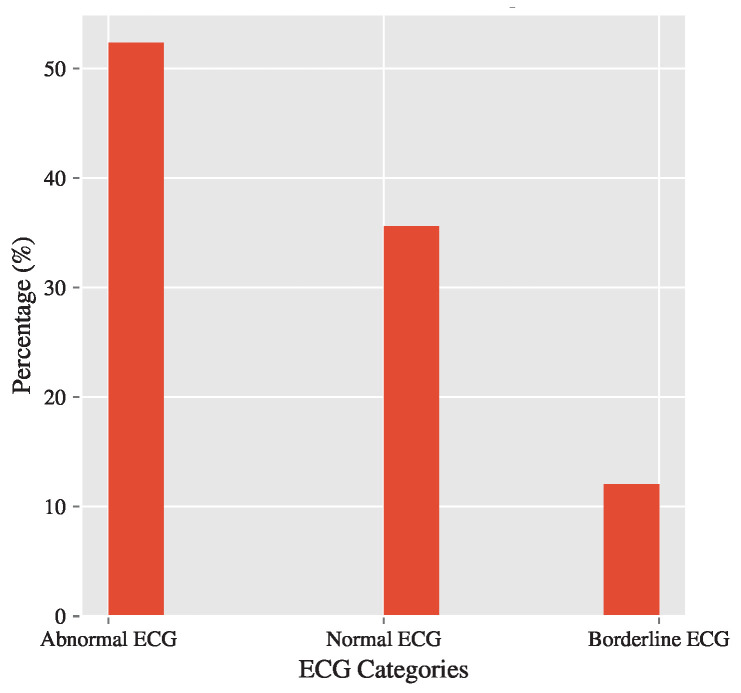
Frequency distribution of ECG diagnosis categories in the dataset ‘DIAGNOSIS’ column, illustrating the proportion of samples classified as ‘Abnormal ECG’, ‘Normal ECG’, and ‘Borderline ECG’. The chart highlights a significant class imbalance, with ‘Abnormal ECG’ constituting the majority of the dataset, emphasizing the challenges this imbalance poses for effective model training and classification.

**Figure 2 diagnostics-15-00903-f002:**
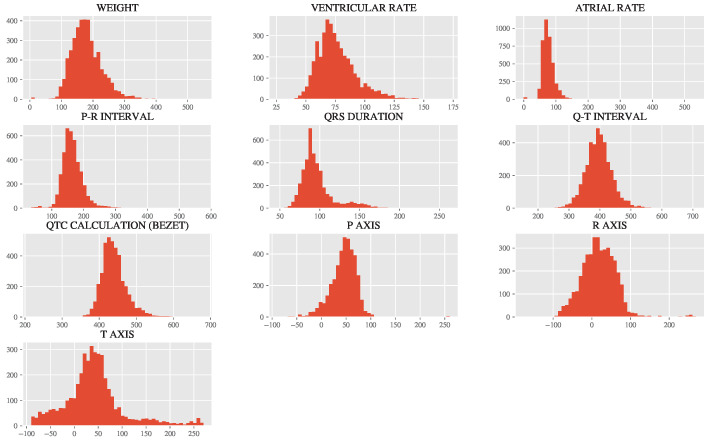
Distribution of ECG-related numerical features before preprocessing, including clinical parameters such as weight, ventricular rate, atrial rate, P-R interval, QRS duration, Q-T interval, and axes (P, R, T). Most features exhibit a Gaussian distribution, while others display long right-tailed distributions. These visualizations guided preprocessing steps, such as outlier analysis and standardization, to optimize the dataset for machine learning model training.

**Figure 3 diagnostics-15-00903-f003:**
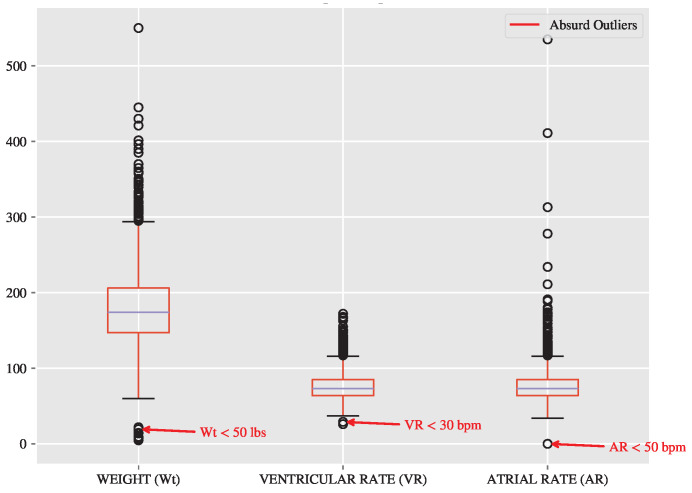
Boxplots of selected numerical features (weight, ventricular rate, and atrial rate) illustrating the results of outlier analysis. The analysis highlights the presence of extreme and unrealistic outliers, such as weights below 50 lbs, ventricular rates below 30 bpm, and atrial rates below 50 bpm. These outliers likely result from errors during the data collection process. The findings informed preprocessing steps, which included the removal of implausible values to improve data quality and model performance.

**Figure 4 diagnostics-15-00903-f004:**
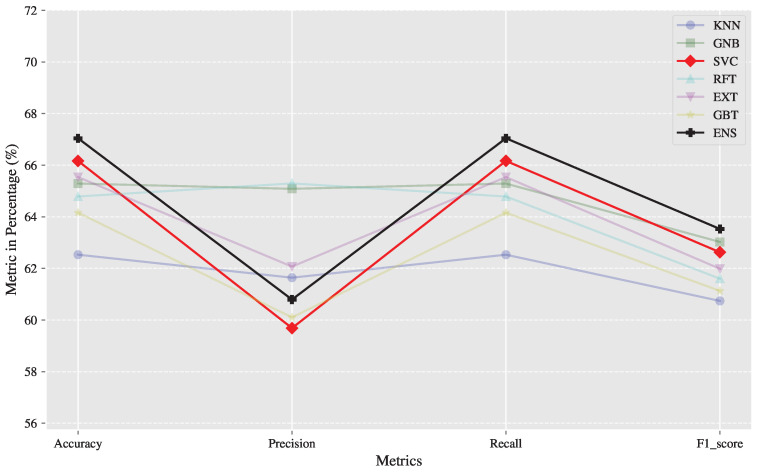
The results in Table 3 of all models evaluated on the test set visualized to compare which model outperforms on all or most of the metrics.

**Figure 5 diagnostics-15-00903-f005:**
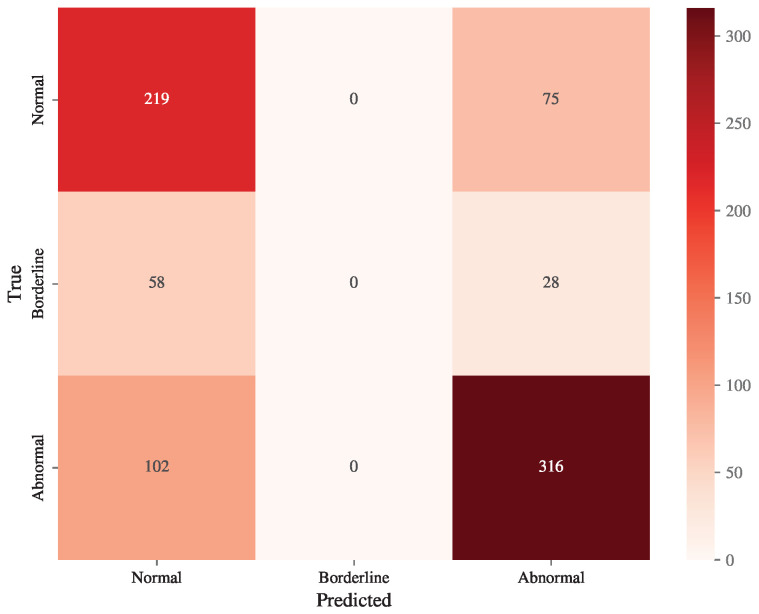
The confusion matrix of predictions of the SVM illustrates the distribution of true versus predicted classes for normal, borderline, and abnormal ECG diagnoses. Borderline ECG predictions were the least frequent, with most misclassifications occurring between borderline and the other two categories. The darker diagonal cells indicate higher accuracy for normal and abnormal ECG classifications, whereas the lighter shading in the borderline column highlights the model’s challenge in correctly identifying this category.

**Figure 6 diagnostics-15-00903-f006:**
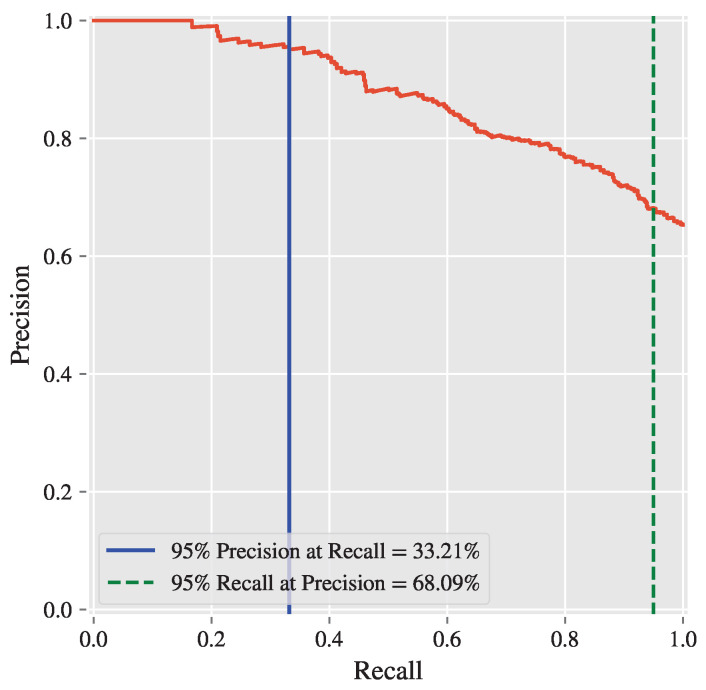
The PR curve highlights the trade-off between precision and recall across varying decision thresholds. Key points include a threshold yielding 95% precision (with recall at 33.21%) and another yielding 95% recall (with precision at 68.09%).

**Figure 7 diagnostics-15-00903-f007:**
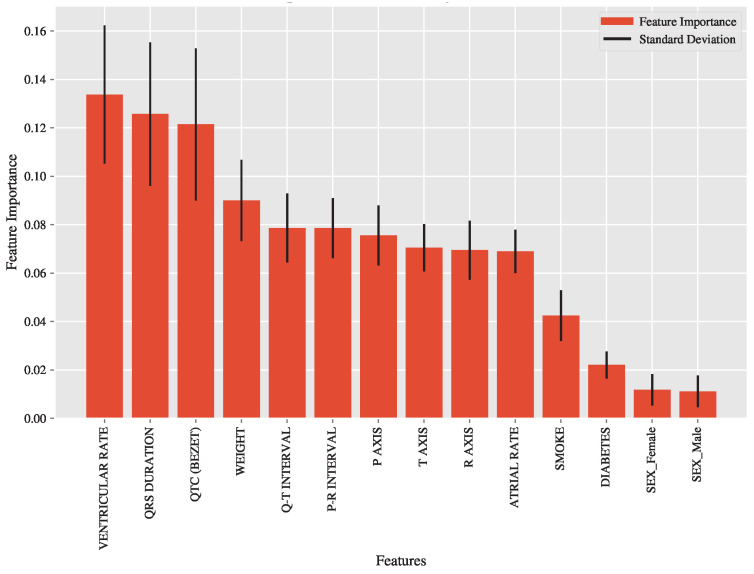
The bar heights represent the relative importance scores, indicating that features such as ventricular rate, QRS duration, and QTC (Bezet) are among the most influential in determining ECG classifications. The error bars illustrate the standard deviation of the importance scores across out-of-bag samples, reflecting variability in feature contributions.

**Table 1 diagnostics-15-00903-t001:** Sample rows from the original patient dataset showcasing ECG-related biometric data and clinical parameters. The table includes features such as sex, weight, smoking status, ventricular rate, atrial rate, and key ECG measurements (e.g., P-R interval, QRS duration, Q-T interval, and QTC (Bezet)), along with target labels for ECG diagnosis (normal, borderline, or abnormal ECG). These raw data entries highlight the diversity of patient characteristics and serve as the foundation for preprocessing and machine learning model training.

SEX	WEIGHT	DIABETES	SMOKE	VENT. RATE	ATRIAL RATE	P-R INTERVAL	QRS DURATION	Q-T INTERVAL	QTC (BEZET)	P AXIS	R AXIS	T AXIS	DIAGNOSIS_LINE_1	...	DIAGNOSIS_LINE_14
Male	204.59	Y	Former	74	74	266.0	88	438	486	37.0	−28	27.0	Abnormal ECG	...	NaN
Male	162.48	N	Never	58	58	135.0	92	410	403	45.0	28	29.0	Normal ECG	...	NaN
Female	119.00	N	Never	75	62	172.0	146	432	482	49.0	−21	118.0	Abnormal ECG	...	Synus Rhythm

**Table 2 diagnostics-15-00903-t002:** Preprocessed patient dataset highlighting key features used for machine learning model training and evaluation. The table includes standardized and encoded numerical features (e.g., P-R interval, QRS duration, Q-T interval, QTC (Bezet), ventricular rate, and weight), as well as one-hot encoded categorical features (e.g., sex and diabetes).

SMOKE	P-R INTERVAL	QRS DURATION	Q-T INTERVAL	QTC (BEZET)	SEX_Female	SEX_Male	DIABETES	ATRIAL RATE	P AXIS	R AXIS	T AXIS	VENT. RATE	WEIGHT
3	1.24	−0.59	−0.38	0.00	1	0	1.00	1.36	0.35	0.32	0.52	0.21	4.23
0	0.50	−1.12	0.10	−1.28	0	1	1.00	−0.22	−0.85	−0.77	−2.32	−0.12	−0.30
0	0.57	0.53	−0.91	0.91	0	1	0.00	1.00	1.75	1.59	0.77	0.72	4.11

**Table 3 diagnostics-15-00903-t003:** The table compares the performance of Gaussian Naive Bayes (GNB), support vector machines (SVM), random forest trees (RFT), extremely randomized trees (EXT), gradient boosted trees (GBT), K-nearest neighbors (KNN), and the ensemble (ENS) consisting of GNB, GBT, EXT, & SVM on the testing dataset. Metrics include testing accuracy, precision, recall, and F1-score. The EXT model achieved the highest overall performance, demonstrating its robustness in handling class imbalance and optimizing the bias-variance trade-off.

Model	Testing Accuracy	Testing Precision	Testing Recall	Testing F1-Score
GNB	65.29%	65.08%	65.29%	63.03%
SVM	66.17%	59.69%	66.17%	62.63%
RFT	64.79%	65.29%	64.79%	61.60%
EXT	65.54%	62.07%	65.54%	61.99%
GBT	64.16%	60.10%	64.16%	61.13%
KNN	62.53%	61.64%	62.53%	60.74%
ENS	67.04%	60.79%	67.04%	63.53%

## Data Availability

The raw data supporting the conclusions of this article will be made available by the authors on request.

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
