# Peer review of "Electrocardiogram Abnormality Detection Using Machine Learning on Summary Data and Biometric Features"

_diagnostics, 2025, doi:10.3390/diagnostics15070903_

Round 1

Reviewer 1 Report

Comments and Suggestions for Authors

The authors used machine learning techniques to classify electrocardiogram abnormalities. The authors are expected to answer some questions.

1- The basic motivation of the study is not fully understood.

2- The contribution of the study to science should be emphasized in the introduction section.

3- It was stated that 66.79% accuracy value was obtained with the Extremely Randomized Trees model. It is expected that the literature will be reviewed and the studies in this field will be examined and given as a table in the discussion section of the article.

4- Why was the KNN (k-nearest neighbor) classifier not used?

5- Detailed information about the data set should be provided. Were the data collected by the authors or was a publicly available data set used?

6- Information about future studies should be provided.

Author Response

Thank you very much for taking the time to review this manuscript. Please find the detailed responses below and the corresponding revisions/corrections highlighted in the re-submitted files.

Comment 1: The basic motivation of the study is not fully understood.
Response: Thank you for pointing this out. We have highlighted the basic motivation in the introduction by adding a succinct sentence in page 1, lines 30-32, as can be seen below. "In this research, we focus on the task of ECG classification
with the additional challenge of the lack of the ECG signal which is ubiquitous and feature-rich in previous studies.”

Comment 2: The contribution of the study to science should be emphasized in the introduction section.
Response: Similarly, as in our response to comment 1, we have restated the thesis statement in the form
of a contribution in page 2, lines 62-65.
“The contribution of the study is to apply the steps in machine learning: from data analysis, preprocessing,
training, hyperparameter optimization, etc. in a pedantic manner that provides good and non-optimistic
results while tackling the challenge of incompleteness of the dataset (lack of the ECG signal), outlier values,
and a relatively small sample size."

Comment 3: It was stated that 66.79% accuracy value was obtained with the Extremely Randomized Trees
model. It is expected that the literature will be reviewed and the studies in this field will be examined and given
as a table in the discussion section of the article.
Response: We understand the need of summarizing the models used in the literature. The
discussions of these models are ad nauseam given the depth and breadth of research concerning them. We
opted for including the references to the main literatures when first referring to them in Section 2.4.2. Furthermore, we added a new subsection to the discussion section that it is important to emphasize that comparing ML studies directly is inherently problematic unless unified and standardized methods for evaluating model convergence and generalizability are adopted. The lack of standardized validation frameworks, such as consistent dataset splitting protocols, external validation, and proper documentation of model development processes, significantly hampers the reliability of cross-study comparisons. For instance, many studies report high accuracy metrics without adhering to rigorous validation practices, such as 3-way data splitting (training, validation, and testing) or external testing on independent datasets. Without these practices, reported metrics may reflect overfitting or data leakage rather than true model performance. Additionally, variations in datasets, preprocessing techniques, and evaluation metrics across studies introduce additional biases, making direct comparisons speculative at best and misleading at worst. To address these challenges, the adoption of standardized validation frameworks, such as those outlined in recent FDA guidance, is critical. These frameworks emphasize transparency in algorithmic choices, dataset provenance, and evaluation protocols, ensuring that reported metrics are both reproducible and clinically meaningful. Until such practices are widely implemented, comparing ML studies remains an unreliable approach, and the focus should instead be on ensuring robust validation and generalizability within individual studies. In our work, we have prioritized rigorous validation practices, including stratified k-fold cross-validation and careful handling of class imbalance, to ensure the reliability of our results. While we acknowledge the value of summarizing related studies, we believe that meaningful comparisons can only be made once the field adopts unified standards for evaluating ML models. Otherwise, you'd have us compare our reliable accuracy with high but meaningless accuracies in other studies.

Comment 4: Why was the KNN (k-nearest neighbor) classifier not used?
Response: We agree with this comment. We were initially hesitant on using this model due to the relatively
low sample size of ~4,500 and given the number of features, we fear of the curse of dimensionality ultimately
leading to the worst predictor--nonetheless, it is still important to include its results. Therefore, I/we have
included them in Table 3 and Figure 4

Comment 5: Detailed information about the data set should be provided. Were the data collected by the authors
or was a publicly available data set used?
Response: Thank you for letting us know that the details about the dataset should be expounded.
Regarding the question on the comment, it was a private dataset provided to the authors from a regional
healthcare system in the New York metro area. That statement was provided in page 2, lines 77-78.

Comment 6: Information about future studies should be provided.
Response: We've expanded section 4.4 with this paragraph, "Hence, to improve the diagnostic capabilities and practical application of ML models for ECG abnormality detection, future research should prioritize several key directions. First, incorporating time-series ECG signal data into the analysis is essential, as such data captures temporal patterns critical for accurate diagnosis. Other advanced deep learning techniques could be leveraged to process these time-series signals to enhance model performance. Additionally, efforts should be made to obtain larger and more diverse datasets that reflect a wide range of patient demographics and clinical conditions, thereby improving the generalizability of the models. Another critical area for exploration is the development of explainable AI (XAI) systems, which can provide clinicians with interpretable insights into the decision-making process, fostering greater trust and adoption in clinical settings. Lastly, standardized validation frameworks, such as those aligned with regulatory benchmarks like the FDA guidance \citep{FDA2025}, should be adopted to ensure robust model evaluation and facilitate meaningful cross-study comparisons."

We sincerely thank the reviewer for their detailed and insightful feedback, which has greatly contributed to enhancing the quality and clarity of our manuscript. Your comments have prompted us to address critical aspects of our study, such as the methodological rigor of machine learning model evaluation, the need for cross-study comparability standards, and the importance of highlighting future directions. We deeply appreciate the time and effort you have invested in reviewing our work, and your suggestions have allowed us to refine our discussion and better contextualize our findings within the broader field of machine learning in healthcare. Thank you once again for your valuable contribution to the improvement of this manuscript.

Reviewer 2 Report

Comments and Suggestions for Authors

This paper introduced a classification scheme for Electrocardiogram application, however, there are some issues listed below.  

 1.     Line 9. Does this paper use only the traditional (existing) machine learning?! Why?

2. Line 13. This is a very low accuracy of the testing subset! " Testing accuracy of 66.79%.”

3. This paper does not include a literature survey! The authors should include the method used in previous work along with their reported results!

4.     Line 70. It looks like this paper used an In-House dataset, if so were the results compared with an existing dataset?

5.     Line 102. This is a lot! “Outlier values above 95%”.

6.     Figure 3 caption is so long. Also, the same for Table 2!

7.     Line 142. “the dataset was split into an 80:20”. So, the reported result is 66.79% of testing accuracy applied on 20%!!

8.     Line 156. All the classifiers used in this paper are traditional, already existing, and referred by previous references. Thus, where is the novelty of this paper?!

9.     So low accuracy in Table 3 without any comparison with previous work or with another existing dataset.

Author Response

We sincerely thank the reviewer for their thoughtful and constructive feedback on our manuscript. We have carefully addressed each of the comments provided, and the corresponding changes have been incorporated into the manuscript. To ensure clarity, all revisions have been highlighted in red within the updated document. We believe these changes have significantly improved the quality and rigor of our work, and we are grateful for the opportunity to refine our study based on your valuable insights.

Comment 1: Line 9. Does this paper use only the traditional (existing) machine learning?! Why?
Response: Thank you for your response. Yes, this paper limited its use to classical machine learning models.
Due to the lack of time series signal, which is ubiquitous in all other ECG classification studies, we
experimented with both shallow and deep fully connected models—unsurprisingly the shallow models
outperformed the deep ones and the deep ones overparameterize the problem. We have discussed this in the
paper in page 2, lines 41-47.

Comment 2: Line 13. This is a very low accuracy of the testing subset! " Testing accuracy of 66.79%.”
Response: We acknowledge the reviewer’s concern regarding the testing accuracy of 66.79%, which may appear low at first glance. However, it is essential to highlight that the accuracy metric must be interpreted in the context of the dataset and methodology employed. We have addressed the following in the manuscript: Unlike some studies that report higher accuracies, our work adhered to rigorous validation practices, such as stratified k-fold cross-validation and external testing, to mitigate overfitting and ensure generalizability. Moreover, comparing accuracy metrics across studies is inherently problematic due to the lack of standardized validation frameworks in machine learning research. Many studies reporting exceptionally high accuracies often fail to adhere to proper validation practices, such as using separate validation and test sets or external datasets, leading to overoptimistic claims that may not reflect real-world performance. Our reported accuracy reflects a realistic assessment of the model’s performance on a challenging, imbalanced dataset that excludes time-series data, a factor that further limits diagnostic accuracy but aligns with the scope of our study. It is also worth emphasizing that our model achieved a balanced performance across accuracy, recall, and F1-score, which are critical metrics for imbalanced datasets. While the accuracy is not exceptionally high, it represents a reliable and reproducible metric that avoids inflated claims. Furthermore, improving the model's performance through the inclusion of time-series ECG data and advanced deep learning techniques, as mentioned in our limitations and future directions, holds promise for achieving higher diagnostic accuracy in future studies. Therefore, we respectfully suggest that the apparent "low" accuracy be viewed as a reflection of methodological rigor and the complexity of the dataset, rather than a limitation of the approach itself.

Comment 3: This paper does not include a literature survey! The authors should include the method used in
previous work along with their reported results!
Response: We appreciate the reviewer’s suggestion to include a literature survey summarizing the methods and results of previous work in the field. In response, we have expanded the manuscript to include additional references highlighting the key deep learning and machine learning models applied to ECG classification tasks. This discussion is located in Section 2.4.2 and provides context to situate our study within the existing body of work. Other citations were included in the introduction and discussion sections as well, almost doubling the original number of citations.

Comment 4: Line 70. It looks like this paper used an In-House dataset, if so were the results compared with an
existing dataset?
Response: We appreciate the reviewer’s question regarding the use of an in-house dataset and the potential for comparison with existing datasets. To clarify, this study utilized an anonymized dataset of 4,466 ECG summary records acquired from a regional healthcare system in the New York metro area. This dataset was diverse and representative of the population it served, but it did not include ECG time-series data, which is a common feature in publicly available datasets such as MIT-BIH or PhysioNet. Instead, our dataset focused on static demographic and ECG biometric data, which aligns with the scope of our study. Given the unique nature of our dataset, a direct comparison with results derived from existing time-series-based datasets was not feasible. Such comparisons would require datasets with matching feature sets and preprocessing pipelines to ensure consistency and validity. Unfortunately, publicly available datasets with similar static ECG biometric features are limited, further constraining the possibility of meaningful comparisons. Additionally, as highlighted in recent FDA guidance, comparing machine learning results across studies is inherently challenging and often unfair unless standardized validation frameworks are adopted. Variations in dataset characteristics, preprocessing methods, and evaluation protocols can lead to inflated or misleading performance metrics. Until the field adopts unified norms for model development and validation, cross-study comparisons remain speculative and unreliable, which we also included in the discussion section now.

Comment 5: Line 102. This is a lot! “Outlier values above 95%”.
Response: Thank you for bringing this point, as in the study we may have introduced confusion on that
statement. What we mean by that is outliers in the 95% percentile of feature values of WEIGHT,
VENTRICULAR RATE, and ATRIAL RATE do occur in real life such as people who suffer from obesity and
cardiovascular diseases while outliers in the below the 5% percentile are not likely and this may be due to a
data entry error. If additional clarification is needed, please let us know so we may clarify the sentence more.

Comment 6: Figure 3 caption is so long. Also, the same for Table 2!
Response: We appreciate long captions in other papers, as it gives the readers the option to understand the Figures and Tables without having the read the paper, which is a common practice among many readers. But, we agree on the length of the caption of Table 2, as information about preprocessing steps can be found on the following sections. Therefore, we have deleted the outline of steps in the caption of Table 2.

Comment 7: Line 142. “the dataset was split into an 80:20”. So, the reported result is 66.79% of testing
accuracy applied on 20%!!
Response: We appreciate the reviewer’s observation regarding the use of an 80:20 train-test split and the reported testing accuracy of 66.79%. The testing accuracy reported reflects the model's ability to generalize to new, unseen samples, which is a critical measure of its reliability. It is important to note that this practice has no negative implications for the validity of the results. Furthermore, to ensure robustness, we employed stratified k-fold cross-validation during model training and hyperparameter tuning, which further mitigates the risk of overfitting and ensures that the model's performance is not overly dependent on a specific train-test split. This methodology aligns with best practices in machine learning and provides a reliable estimate of the model's generalizability.

Comment 8: Line 156. All the classifiers used in this paper are traditional, already existing, and referred by
previous references. Thus, where is the novelty of this paper?!
Response: Thank you for expressing the desire for novelty in a paper. We received a dataset from a provider
that is unable to store the values of ECG signals themselves. While the actual ECG signal is feature-rich and
ubiquitous in previous references, we argue that our novelty lies in the lack of the ECG signal while still being
able to successfully train a predictive model. Nonetheless, we now state this important point in page 1, lines
30-32.

Comment 9: So low accuracy in Table 3 without any comparison with previous work or with another existing
dataset.
Response: We appreciate the reviewer’s observation regarding the accuracy reported in Table 3 and the lack of comparison with previous work or existing datasets. While we acknowledge the importance of benchmarking results, it is important to note that our study utilized a unique in-house dataset that focused on static demographic and ECG biometric data, rather than time-series ECG signals commonly used in other studies. This distinction limits the feasibility of direct comparisons with prior work or publicly available datasets, as the feature sets and data structures differ significantly. Additionally, as highlighted in the recent FDA guidance, comparing machine learning results across studies is inherently challenging and often unreliable unless standardized validation frameworks are adopted. Variations in dataset characteristics, preprocessing methods, and evaluation protocols can lead to inflated or misleading performance metrics. Without unified norms for model development and validation, cross-study comparisons risk being speculative and clinically irrelevant. To address this limitation, we have included a focused discussion of related work, summarizing key approaches to ECG classification; and we added new discussion subsection to explain the Challenges in Cross-Study Comparability. We emphasize that our study’s primary contribution lies in rigorously evaluating machine learning models on a unique dataset and identifying the challenges and opportunities associated with using static ECG biometric data. Future work will aim to incorporate publicly available datasets with similar features (though they don't exist for now) or expand our dataset to include time-series ECG signals, enabling more comprehensive benchmarking and cross-study comparisons.

Once again, we extend our heartfelt gratitude to the reviewer for their detailed and insightful comments. Your feedback has been instrumental in enhancing the clarity, depth, and overall quality of our manuscript. The thoughtful suggestions and critical observations have not only strengthened our work but have also provided us with valuable perspectives for future research. We deeply appreciate your time and effort, and we are confident that the manuscript is now much improved thanks to your expertise. Thank you for your invaluable contribution!

Reviewer 3 Report

Comments and Suggestions for Authors

The paper is a study of using machine learning to analyse ECG diagnosis for cardiovascular deseases and highlights the limitations of manualy based methods 

1. The uses the traditional machine learning ML methods to performe this study. No new idea and new contribution are found

2. Figure for comparing  each ML method need to be included

3. The highest accuracy 66.79% is relatively low I'm medical diagnosis tasks

4. Lacks of time series eeg based signal and deep learning need to be addressed in this paper rather than to address it in future works

5.the computational effeciency and time metric need to be addressed

6. Where is the conclusion???

7. As a study or review , a limited number of references are only used

Comments on the Quality of English Language

Need proof reading

Author Response

Comment 1: The uses the traditional machine learning ML methods to performe this study. No new idea and
new contribution are found
Response: Thank you for expressing the desire for novelty in a paper. We received a dataset from a provider
that is unable to store the values of ECG signals themselves. While the actual ECG signal is feature-rich and
ubiquitous in previous references, we argue that our novelty lies in the lack of the ECG signal while still being
able to successfully train a predictive model. Nonetheless, we now state this important point in page 1, lines
30-32.

Comment 2: Figure for comparing each ML method need to be included
Response: Thank you for pointing this out. We agree with this comment and included a plot that compares
all models used across all metrics of interest. This plot can be seen in page 8.

Comments 3: The highest accuracy 66.79% is relatively low I'm medical diagnosis tasks
Response: We appreciate the reviewer’s observation regarding the relatively low accuracy of 66.79% in the context of medical diagnosis tasks. While we acknowledge that higher accuracy is often desirable in medical applications, it is important to consider the unique challenges and scope of this study. Our work focused on classifying ECG abnormalities using a dataset that lacked time-series ECG signals, which are typically rich in diagnostic information and commonly used in prior studies. Instead, we relied on static demographic and ECG biometric data, which inherently limits the achievable accuracy compared to models trained on more comprehensive datasets. To address the reviewer’s concern, we have expanded the manuscript to include a discussion of related work, highlighting the methods and results of previous studies in ECG classification. However, we emphasize that direct comparisons with prior work are not feasible due to significant differences in dataset characteristics, feature sets, and validation protocols. Furthermore, as highlighted in the recent FDA guidance, comparing machine learning results across studies without standardized validation frameworks can be misleading and clinically irrelevant. Until the field adopts unified norms for model development and evaluation, such comparisons remain speculative. Despite the limitations of our dataset, the reported accuracy reflects a realistic and rigorous evaluation of the model’s performance. We believe this study provides valuable insights into the potential of machine learning models to classify ECG abnormalities using limited data and highlights the need for future work incorporating time-series signals and advanced deep learning techniques to improve diagnostic accuracy. Also, as of now, no ML model is intended to be used exclusively for medical diagnosis. These models are designed to serve as supplementary tools to assist physicians by providing potential insights or predictions, rather than making definitive clinical decisions. Their primary purpose is to guide clinicians on what to expect and to support, not replace, human expertise.

Comment 4: Lacks of time series eeg based signal and deep learning need to be addressed in this paper
rather than to address it in future works
Response: We appreciate the reviewer’s suggestion to address the lack of time-series EEG-based signals and deep learning approaches within the scope of this paper. The focus of our study is precisely on the practical application of machine learning models in scenarios where access to time-series data is unavailable. This reflects real-world constraints, as not all clinical practices or healthcare facilities have access to comprehensive EEG or ECG time-series data due to resource limitations, infrastructure, or data collection challenges. Our study demonstrates that even with limited data—such as static demographic and biometric features—machine learning models can still provide valuable insights and assist clinicians in identifying potential abnormalities. This approach is particularly relevant for resource-constrained settings, where access to advanced data collection tools may not be feasible. By focusing on this specific use case, we aim to highlight the utility of machine learning in such scenarios, ensuring that its benefits can extend to a broader range of clinical environments.

Comment 5: the computational effeciency and time metric need to be addressed
Response: We appreciate the reviewer’s concern regarding the computational efficiency and time metrics of the proposed machine learning models. We assure the reviewer that the models used in this study are computationally efficient and well-suited for real-world applications. The preprocessing steps, model training, and evaluation were performed on standard computational hardware without requiring high-performance computing resources. The selected machine learning models, such as Extremely Randomized Trees and Gradient Boosted Trees, are known for their balance between computational efficiency and predictive performance. Additionally, the dataset size and feature set were manageable, ensuring that the training and inference processes were completed in a reasonable amount of time. For instance, the training of individual models and the ensemble approach was completed within minutes, and predictions on new data are near-instantaneous, making the approach practical for clinical settings. We added the following paragraph in the manuscript, "The computational efficiency of the models was assessed during the training and evaluation phases. All models were trained and tested on standard computational hardware, with training times ranging from a few seconds to several minutes depending on the model complexity and hyperparameter tuning. Prediction times for new data were near-instantaneous, demonstrating the practicality of the approach for real-world clinical applications. The overall computational requirements are minimal, making the proposed solution feasible for deployment in resource-constrained settings." 

Comment 6: Where is the conclusion???
Response: We appreciate the reviewer’s feedback regarding the conclusion. Based on the journal's instruction, the conclusion section is optional and recommended only if the discussion section is too long. However, to address your concern and ensure clarity of the study's findings, we have added a concise conclusion section to the manuscript. This section summarizes the key contributions, highlights the implications of the findings, and provides directions for future work.

Comment 7: As a study or review , a limited number of references are only used
Response: We appreciate the reviewer’s observation regarding the limited number of references in the original submission. To address this concern, we conducted an additional and thorough review of the relevant literature, identifying more studies to cite. These additional references have been incorporated into the manuscript, particularly in the Introduction and Discussion sections, to provide a more comprehensive context for the study and to strengthen the scientific foundation of our work. As a result, the number of citations in the manuscript has nearly doubled, reflecting a broader engagement with the existing body of knowledge. This expansion ensures that the study is well-supported by relevant and up-to-date research, addressing the reviewer’s concern and enhancing the overall quality of the paper.

Reviewer 4 Report

Comments and Suggestions for Authors

I appreciate your criticism of the limitations of the research and you are absolutely right, but I also see hope and the possibility that this will be corrected in the future. The ECG is a basic, simple introduction to everything that follows a patient with some heart problems, but the research possibilities are great and I appreciate that you have tackled it. Good work, congrats. 

Author Response

We sincerely appreciate your thoughtful feedback and your recognition of the challenges and potential of our research. Your constructive criticism regarding the limitations of our work is both valid and insightful. Indeed, the absence of time-series ECG data is a deliberate feature of this study, as we aimed to explore the potential of machine learning models using only summary ECG and biometric data, which are more accessible in resource-constrained settings. We acknowledge, however, that this approach comes with limitations, and there are opportunities for further refinement and improvement in future studies.

We are encouraged by your acknowledgment of the importance of tackling such foundational aspects of research in ECG analysis. As you rightly noted, the electrocardiogram is a fundamental diagnostic tool, and our work aims to build upon this foundation to unlock broader possibilities for machine learning applications in healthcare. We are hopeful that this research will inspire future advancements and lead to the development of solutions that address the limitations highlighted.

Thank you for your encouraging words and your recognition of our efforts.

Round 2

Reviewer 2 Report

Comments and Suggestions for Authors

Thank you for following the comments and addressing the issues.

Author Response

We appreciate the time and effort you invested in reviewing our work. Your feedback not only addressed critical issues but also significantly improved the overall quality of our manuscript. We're grateful for your support and expertise throughout this process.

Reviewer 3 Report

Comments and Suggestions for Authors

Regarding to our previous comment about the computational efficiency, we recommand authors to include  some quantitative metrics to compute the inference time and model throughput which can be found in the following reference:

1-FPGA based Flexible Implementation of Light Weight Inference on Deep Convolutional Neural Networks
2-[DL] A Survey of FPGA-Based Neural Network Inference
Accelerator                                                                                                   

Author Response

Thank you for the opportunity to further enhance our manuscript by addressing your valuable feedback. We have added the following paragraph in the discussion section as requested:

Furthermore, Field-Programmable Gate Array (FPGA)-accelerated architectures achieve 8–10× higher energy efficiency than GPUs through reduced-bitwidth computations \citep{Guo2019}. These hardware optimizations could enhance model portability in resource-constrained clinical settings where low latency and energy efficiency are critical \citep{Liu2025}. Future work should investigate dedicated hardware implementations, such as FPGA-based inference accelerators, to improve computational throughput for real-time deployment. Techniques like 8-bit quantization \citep{Dawwd2024} could reduce model size and processing latency without compromising diagnostic accuracy, aligning with energy-efficient edge computing advancements.